# Etiological and epidemiological characteristics of severe acute respiratory infection caused by multiple viruses and *Mycoplasma pneumoniae* in adult patients in Jinshan, Shanghai: A pilot hospital-based surveillance study

**Jian Li** [1]ʘ*, **Can-Lei Song**[2]ʘ, **Tang Wang**[2]ʘ, **Yu-Long Ye**[3], **Jian-Ru Du**[3], **Shu-Hua Li**[2], **Jian-Min Zhu**[2]*

1 Clinical Research Center, Ruijin Hospital, Shanghai Jiao Tong University School of Medicine, Shanghai, China, 2 Department of Acute Infectious Diseases Control, Jinshan District Center for Diseases Control and Prevention, Shanghai, China, 3 Department of Microbiology, Jinshan District Center for Diseases Control and Prevention, Shanghai, China

ʘ These authors contributed equally to this work.
* nclijian@163.com (JL); zhujm12@163.com (JMZ)

## Abstract

### Background

Severe acute respiratory infection (SARI) results in a tremendous disease burden worldwide. Available research on active surveillance among hospitalized adult patients suffering from SARI in China is limited. This pilot study aimed to identify associated etiologies and describe the demographic, epidemiological and clinical profiles of hospitalized SARI patients aged over 16 years in Jinshan, Shanghai.

### Methods

Active surveillance was conducted at 1 sentinel hospital in Jinshan district, Shanghai, from April 2017 to March 2018. Hospitalized SARI patients aged over 16 years old were enrolled, and nasopharyngeal swabs were collected within 24 hours of admission and tested for multiple respiratory viruses (including 18 common viruses) and *Mycoplasma pneumoniae* with real-time polymerase chain reaction. Demographic, epidemiological and clinical information was obtained from case report forms.

### Results

In total, 397 SARI patients were enrolled; the median age was 68 years, and 194 (48.9%) patients were male. A total of 278 (70.0%) patients had at least one underlying chronic medical condition. The most frequent symptoms were cough (99.2%) and sputum production (88.4%). The median duration of hospitalization was 10 days. A total of 250 infection

**Data Availability Statement:** All relevant data are contained within the paper and its Supporting Information files.

**Funding:** This work was supported by the Research Project of Shanghai Municipal Health Commission (201940428) for Can-Lei Song and the Infectious Disease and Epidemiology Project of the 6th Jinshan District Medical Key Specialty Construction (JSZK2019B05) for Shu-Hua Li. The funder had no role in study design, data collection and analysis, decision to publish or preparation of manuscript.

**Competing interests:** The authors have declared that no competing interests exist.

patients (63.0%) were positive for at least one pathogen, of whom 198 (49.9%) were positive for a single pathogen and 52 (13.1%) were positive for multiple pathogens. The pathogens identified most frequently were *M. pneumoniae* (23.9%, 95/397), followed by adenovirus (AdV) (11.6%, 46/397), influenza virus A/H3N2 (Flu A/H3N2) (11.1%, 44/397), human rhinovirus (HRhV) (8.1%, 32/397), influenza virus B/Yamagata (Flu B/Yamagata) (6.3%, 25/397), pandemic influenza virus A/H1N1 (Flu A/pH1N1) (4.0%, 16/397), parainfluenza virus (PIV) type 1 (2.0%, 8/397), human coronavirus (HCoV) type NL63 (2.0%, 8/397), HCoV 229E (1.5%, 6/397), HCoV HKU1 (1.5%, 6/397), PIV 3 (1.5%, 6/397), human metapneumovirus (HMPV) (1.5%, 6/397), PIV 4 (1.3%, 5/397), HCoV OC43 (1.0%, 4/397), influenza virus B/Victoria (Flu B/Victoria) (0.5%, 2/397), respiratory syncytial virus (RSV) type B (0.5%, 2/397), and human bocavirus (HBoV) (0.3%, 1/397). The seasonality of pathogen-confirmed SARI patients had a bimodal distribution, with the first peak in the summer and the second peak in the winter. Statistically significant differences were observed with respect to the rates of dyspnea, radiographically diagnosed pneumonia and the presence of at least one comorbidity in patients who were infected with only *M. pneumoniae*, AdV, HRhV, Flu A/H3N2, Flu A /pH1N1 or Flu B/Yamagata. The differences in the positivity rates of the above 6 pathogens among the different age groups were nonsignificant.

## Conclusions

*M. pneumoniae*, AdV and Flu A/H3N2 were the main pathogens detected in hospitalized SARI patients aged over 16 years old in Jinshan district, Shanghai. Our findings highlight the importance of sustained multipathogen surveillance among SARI patients in sentinel hospitals, which can provide useful information on SARI etiologies, epidemiology, and clinical characteristics.

## Background

Severe acute respiratory infection (SARI) has been considered an important contributor to morbidity and mortality in all age groups, particularly children, elderly individuals and individuals with compromised immune, cardiac and pulmonary systems, worldwide [1–3]. It is estimated that SARI causes approximately 4.2 million deaths annually. Of these, up to 90% are believed to occur in developing countries [4]. Various viral and bacterial pathogens are associated with SARI. Due to their extremely high potential for human-to-human transmission, these pathogens pose a substantial risk to human health. While bacterial infection has a substantial influence on the development of severe pneumonia [5], a significant proportion of SARIs are attributed to viral pathogens, such as influenza viruses A and B (Flu A/B), parainfluenza viruses (PIV), adenoviruses (AdVs), respiratory syncytial viruses (RSVs), human coronaviruses (HCoVs) and human rhinoviruses (HRhVs) [6]. Nevertheless, owing to the lack of gold standard diagnostic methods to rapidly identify etiological agents, most patients are treated with antibiotics empirically [7]. Rapid etiologic diagnosis therefore remains a significant public health challenge.

Routine pathogen monitoring is critical for preparedness and response to the SARI epidemic. Since SARI is the leading cause of hospitalization in children under the age of 5 years and of febrile episodes in infants younger than 3 months old, most available studies regarding

the burden of SARI focus on viral infections in children [8–11]. A SARI surveillance study in China revealed that 90% of patients were aged <15 years [12]. In addition, the majority of the data on the epidemiology of the etiologic agents of SARI were collected in developed regions. The epidemiological characteristics and distributions of the major viral pathogens in adult SARI patients are still limited in developing regions [13].

*Mycoplasma pneumoniae* has long been considered an important etiology of respiratory disease and is more frequently isolated among children and young adults [14, 15]. Research on active surveillance in hospitalized adult patients suffering from SARI in China is scarce. Accordingly, a pilot study on active surveillance of SARI was initiated to characterize community-acquired pulmonary infections and to monitor the epidemiological and etiologic characteristics of SARI caused by various viral pathogens and *M. pneumoniae* in adult patients in Jinshan district, Shanghai. The aim of the present study was to characterize the demographic and epidemiological characteristics of SARI, identify the etiologies and assess the clinical profiles associated with SARIs in hospitalized adult patients in Jinshan, Shanghai, by performing 12 months of active surveillance from April 2017 to March 2018.

## Materials and methods

### Study setting

Jinshan district is a suburb located in southwest Shanghai, P.R. China. One year active surveillance was initiated at Jinshan District Central Hospital since April 2017. This hospital was selected because it is the largest general hospital in the district and a national surveillance sentinel site for the influenza virus. It serves most of the population in Jinshan district, with a total of 636 beds. In 2017, the registered population in Jinshan district was 523,641, of which 467,320 (89.24%) were adults [16].

### Study subjects

All patients aged over 16 years who were admitted to the intensive care unit, respiratory medicine department and general wards in the hospital were screened by a trained physician between April 2017 and March 2018. Patients were diagnosed with SARI according to the World Health Organization (WHO) definition, which includes acute respiratory infection with a measured fever of $\geq 38$°C, cough onset within the last 10 days and required hospitalization [8].

### Data collection

After hospital admission, a standard case report form was completed for each eligible patient. The form comprised information on demographic characteristics (sex, age, weight, height, residence), vaccination (received an influenza vaccine 1 year before illness onset, ever received a pneumococcal conjugate vaccine), admission diagnosis, comorbidities (asthma, chronic bronchitis, chronic obstructive pulmonary disease (COPD), hypertension, diabetes, cardiovascular disease, tumor), clinical presentation (fever, cough, difficult breathing, sore throat), antibiotic treatments prior to hospitalization, exposure history (smoking, visiting a live poultry market, contact with live poultry, contact with a patient with fever and respiratory symptoms within 2 weeks before illness onset). At discharge, the form was updated to include information about treatment in the hospital, chest computed tomographic (CT) findings, complications and prognosis. Data were collected by the trained physician. To ensure the accuracy of the data, spouses or caregivers who lived with the patients for more than 2 weeks before illness onset were interviewed, and the medical records of the patients were reviewed. Two radiologists

interpreted chest CT scans independently. In the case of a disagreement, a third radiologist was consulted to reach a final decision. All the person-identifiable information of patients was masked during or after data collection.

## Specimen collection and laboratory testing

A single flocked polyester nasopharyngeal swab (Becton Dickinson, USA, MD) sample was collected from each SARI patient by a nurse within 24 hours of admission following a standard procedure. The swab was inserted into a cryovial containing 3 ml of viral transport medium (Tiandz, China, Beijing). The specimens were stored at 4˚C in the hospital and transferred within 24 hours of collection to the laboratory at Jinshan District Center for Disease Control and Prevention (CDC), where they were preserved at -70˚C until used. Viral RNA and DNA were extracted from 200-μl samples using the QIAamp Viral RNA/DNA Mini Kit (Qiagen, Hilden, Germany) following the manufacturer's instructions. To guarantee integrity, specimens were lysed under denaturing conditions to deactivate RNases as per the manufacturer's instructions. RNA and DNA were eluted in 60 μl of low-salt buffer, and impurities were removed. Viral nucleic acid extracts were further processed by multiplex real-time reverse transcription polymerase chain reaction (RT-PCR). The qualitative RespiFinder 2SMART multiplex real-time RT-PCR diagnostic strategy (Geneodx, Shanghai, China) was adopted to detect 15 respiratory pathogens, including PIV (types 1, 2, 3 and 4), HCoV (types 229E, OC43, HKU1 and NL63), RSV (types A and B), HRhV, AdV, human metapneumovirus (HMPV), human bocavirus (HBoV) and *M. pneumoniae*, using the CFX96™ real-time PCR system (Bio-Rad, Hercules, CA, USA) according to the manufacturer's protocols. In addition, RNA from each specimen was identified for specific primers and probes that target Flu A/B using real-time RT-PCR following the US CDC's protocol. Specimens that were positive for Flu A and Flu B were subsequently subtyped for pandemic influenza virus A/H1N1 (Flu A/pH1N1) and seasonal influenza virus A/H3N2 (Flu A/H3N2) and Flu B/Yamagata and Flu B/Victoria, respectively [17]. These tests were performed in the biosafety level 2 laboratory of the Jinshan CDC.

## Statistics

The collected data were double-entered into a database constructed in EpiData 3.1. Logic checks to assess the quality of data entry were conducted. Single infection was defined as infection caused by one pathogen, and multiple infection was defined as infection caused by at least 2 pathogens (virus/virus, virus/*M. pneumoniae*) in a single specimen. Continuous data are reported as medians and interquartile ranges (IQRs), and the Mann-Whitney U test was used to compare differences between groups. Categorical data are expressed as frequencies and proportions, and the chi-squared test or Fisher's exact test, as appropriate, was used to compare patients with and without confirmed pathogens in terms of demographics, clinical characteristics, epidemiologic characteristics, treatment and prognosis. Bonferroni's correction was used for pairwise comparisons. For proportions, the binomial 95% confidence interval is reported. The analysis was performed using SPSS v. 25.0 (IBM Corporation, Armonk, NY, USA), and all tests were two-sided with a 5% significance level.

## Ethics statement

This study was part of a hospital-based SARI surveillance program in Shanghai and was approved by the ethical review committee of the Shanghai Municipal Center for Disease Control and Prevention (Ref #: 2015–14). Written informed consent was obtained from patients or

proxies before enrollment and from parents or guardians of those under 18 years old. This study was conducted in accordance with the Declaration of Helsinki.

## Results

### Demographic characteristics

From April 2017 to March 2018, a total of 397 patients meeting the SARI case definition were admitted to our hospital. One or more pathogens were detected in 250 patients (63.0%; 95% CI: 58.2–67.7%), and negative results were obtained from the remaining 147 patients. The median age of the patients was 68 years (IQR: 59–78; range: 16 to 99 years). Among the SARI patients, 194 (48.9%) were male, and 203 (51.9%) were female. The majority of patients were elderly patients aged 60 or more years (295 cases), accounted for 74.3% of the total patients; 58 (14.6%) patients were 40–59 years of age, and 19 (4.8%) patients were 30–39 years of age. Those less than 30 years old represented only 6.3% of the total patients (25 cases). The percentages of patients with a body mass index (BMI) <20, between 20 and 25, and >25 were 29.7%, 52.4% and 17.9%, respectively. A total of 278 SARI patients (70.0%) had at least one comorbidity, and 119 patients had no comorbidity (Table 1). There were no significant differences in sex, age, BMI and underlying chronic medical conditions between SARI patients with confirmed pathogens and those without confirmed pathogens ($P>0.05$).

**Table 1.  Demographic characteristics of adult SARI patients in a surveillance hospital in Jinshan, Shanghai, April 2017 to March 2018.**

| Characteristics | SARI patients | | | P value* |
|---|---|---|---|---|
| | All (%) [n = 397] | With confirmed pathogens (%) [n = 250] | Without confirmed pathogens (%) [n = 147] | |
| Sex | | | | 0.315 |
| Male | 194(48.9) | 127(50.8) | 67(45.6) | |
| Female | 203(51.1) | 123(49.2) | 80(54.4) | |
| Age group (median, years) | 68.0 | 67.0 | 69.0 | 0.357 |
| <30 | 25(6.3) | 17(6.8) | 8(5.4) | 0.786 |
| 30–39 | 19(4.8) | 10(4.0) | 9(6.1) | |
| 40–59 | 58(14.6) | 39(15.6) | 19(12.9) | |
| 60–79 | 207(52.1) | 128(51.2) | 79(53.7) | |
| ≥80 | 88(22.2) | 56(22.4) | 32(21.9) | |
| BMI | | | | 0.657 |
| <20 | 118(29.7) | 73(29.2) | 45(30.6) | |
| 20–25 | 208(52.4) | 135(54.0) | 73(49/7) | |
| >25 | 71(17.9) | 42(16.8) | 29(19.7) | |
| Chronic medical conditions | | | | |
| At least one | 278(70.0) | 178(71.2) | 100(68.0) | 0.505 |
| Asthma | 12(3.0) | 6(2.4) | 6(4.1) | 0.345 |
| Chronic bronchitis | 49(12.3) | 30(12.0) | 19(12.9) | 0.787 |
| COPD | 28(7.1) | 13(5.2) | 15(10.2) | 0.060 |
| Hypertension | 152(38.3) | 95(38.0) | 57(38.8) | 0.878 |
| Cardiovascular disease | 30(7.6) | 22(8.8) | 8(5.4) | 0.222 |
| Diabetes | 61(15.4) | 38(15.2) | 23(15.6) | 0.905 |
| Cerebrovascular disorder | 20(5.0) | 14(5.6) | 6(4.1) | 0.504 |
| Tumor | 19(4.8) | 14(5.6) | 5(3.4) | 0.322 |

*P values denote comparisons between SARI patients with confirmed pathogens and SARI patients without confirmed pathogens.

## Etiologies

Of the 397 SARI patients, 198 (49.9%; 95% CI: 45.0–54.8%) patients had single infection, while 52 (13.1%; 95% CI: 9.8–16.4%) patients had multiple infection. The most prevalent pathogen identified was *M. pneumoniae* in 95 (23.9% of the total samples) patients, followed by AdV in 46 (11.6%) patients, Flu A/H3N2 in 44 (11.1%) patients, HRhV in 32 (8.1%) patients, Flu B/Yamagata in 25 (6.3%) patients, and Flu A /pH1N1 in 16 (4.0%) patients. Other viruses, including PIV 1, HCoV NL63, HCoV 229E, HCoV HKU1, PIV 3, HMPV, PIV 4, HCoV OC43, Flu B/Victoria, RSV B and HBoV, were detected in a 0.3% to 2.0% of samples (Table 2). The most frequently detected pathogens in patients with multiple infection were *M. pneumoniae* (84.6%, 44/52), AdV (28.8%, 15/52), HRhV (25.0%, 13/52), and Flu A/H3N2 (17.3%, 9/52).

**Table 2. Etiological agent distributions among adult SARI patients in a surveillance hospital in Jinshan, Shanghai, April 2017 to March 2018.**

| Etiological agent | Frequency[#] (n) | Percent of samples[*] (%) |
|---|---|---|
| Influenza virus A | | |
| pH1N1 | 16 | 4.0 |
| H3N2 | 44 | 11.1 |
| Influenza virus B | | |
| Yamagata | 25 | 6.3 |
| Victoria | 2 | 0.5 |
| Parainfluenza virus | | |
| Type 1 | 8 | 2.0 |
| Type 2 | 0 | 0 |
| Type 3 | 6 | 1.5 |
| Type 4 | 5 | 1.3 |
| Human coronavirus | | |
| Type 229E | 6 | 1.5 |
| Type OC43 | 4 | 1.0 |
| Type HKU1 | 6 | 1.5 |
| Type NL63 | 8 | 2.0 |
| Respiratory syncytial virus | | |
| Type A | 0 | 0 |
| Type B | 2 | 0.5 |
| Human rhinovirus | 32 | 8.1 |
| Adenovirus | 46 | 11.6 |
| Human metapneumovirus | 6 | 1.5 |
| Human bocavirus | 1 | 0.3 |
| *Mycoplasma pneumoniae* | 95 | 23.9 |
| Single infection | 198 | 49.9 |
| Multiple infection | | |
| 2 pathogens | 43 | 10.8 |
| 3 pathogens | 8 | 2.0 |
| 4 pathogens | 1 | 0.3 |

[#]The frequency of each pathogen may include both the samples with single infection and those with multiple infection, and their total number is larger than the sum of samples with single infection and multiple infection.
[*]Percent of samples is the frequency of samples with a positive etiology divided by the total enrolled samples (397 cases).

## Clinical and epidemiological characteristics

Pneumonia (222 cases, 55.9%) was the most common clinical diagnosis made by clinicians on admission, followed by bronchiolitis (68 cases, 17.1%). The most common symptoms on admission were cough (99.2%) and sputum production (88.4%), followed by thoracalgia (7.1%) and pharyngalgia (6.8%). Of the 397 SARI patients, a temperature $\geq 39°C$ was recorded in 189 SARI patients (47.6%) on admission. A total of 382 patients (96.2%) underwent chest CT, of whom 258 (67.5%) were reported to have radiographic evidence of pneumonia; the remaining 15 patients did not undergo chest CT examination. Thirty-two SARI patients had exposure to a patient with fever and respiratory symptoms, while 30 SARI patients had contact with live poultry 2 weeks before illness onset. Among the 397 patients, only 5 patients had received a pneumococcal conjugate vaccine, and 1 patient was vaccinated against influenza (Table 3). No significant differences in the proportions of clinical and epidemiological characteristics between SARI patients with confirmed pathogens and those without confirmed pathogens were found, except for chest radiographic examination findings. As illustrated in Table 4, the differences in the proportions of dyspnea, radiographic diagnosis of pneumonia and the presence of at least one comorbidity among patients infected with only one of the 6 main pathogens, including *M. pneumoniae*, AdV, HRhV, Flu A/H3N2, Flu A /pH1N1 and Flu B/Yamagata, were statistically significant. Notably, the proportion of patients with radiographic evidence of pneumonia was the highest in patients infected by *M. pneumoniae* (74.5%), and dyspnea was the most common presentation in patients with HRhV (21.1%).

**Table 3. Clinical and epidemiological characteristics of adult SARI patients in a surveillance hospital in Jinshan, Shanghai, April 2017 to March 2018.**

| Characteristics | SARI patients | | | P value* |
|---|---|---|---|---|
| | All (%) [n = 397] | With confirmed pathogens (%) [n = 250] | Without confirmed pathogens (%) [n = 147] | |
| Temperature ≥39˚C | 189(47.6) | 126(50.4) | 63(42.9) | 0.176 |
| Cough | 394(99.2) | 249(99.6) | 145(98.6) | 0.558 |
| Sputum production | 351(88.4) | 219(87.6) | 132(89.8) | 0.509 |
| Pharyngalgia | 27(6.8) | 18(7.2) | 9(6.1) | 0.680 |
| Thoracalgia | 28(7.1) | 19(7.6) | 9(6.1) | 0.687 |
| Dyspnea | 19(4.8) | 11(4.4) | 8(5.4) | 0.808 |
| Runny nose | 11(2.8) | 7(2.8) | 4(2.7) | 1.000 |
| Vomiting | 15(3.8) | 10(4.0) | 5(3.4) | 0.795 |
| Acceptance of chest radiographic exam | 382(96.2) | 236(94.4) | 146(99.3) | **0.013** |
| Presence of radiographic diagnosis of pneumonia | 258/382(67.5) | 153/236(64.8) | 105/146(71.9) | 0.349 |
| Visited a live poultry market | 3(0.8) | 3(1.2) | 0(0) | 0.299 |
| Contact with live poultry | 30(7.6) | 19(7.6) | 11(7.5) | 1.000 |
| Contact with patient with fever | 32(8.1) | 24(9.6) | 8(5.4) | 0.182 |
| Smoking | | | | 0.860 |
| Current | 43(10.8) | 28(11.2) | 15(10.2) | |
| Former | 66(16.6) | 43(17.2) | 23(15.6) | |
| Never | 288(72.6) | 179(71.6) | 109(74.2) | |
| Vaccinated with pneumococcal conjugate vaccine | 5(1.3) | 3(1.2) | 2(1.4) | 1.000 |
| Vaccinated with influenza vaccine | 1(0.3) | 1(0.4) | 0(0) | 1.000 |

*P values denote comparisons between SARI patients with confirmed pathogens and SARI patients without confirmed pathogens.

**Table 4. Comparison of characteristics of SARI patients infected with only one of the 6 main pathogens in a surveillance hospital in Jinshan, Shanghai, April 2017 to March 2018.**

| Characteristics | *M. pneumoniae*(%) [n = 51] | AdV (%) [n = 31] | HRhV (%) [n = 19] | Flu A/H3N2 (%) [n = 35] | Flu B/Yamagata (%) [n = 21] | Flu A /pH1N1 (%) [n = 16] | P value[*] |
|---|---|---|---|---|---|---|---|
| Sex |  |  |  |  |  |  | 0.750 |
| Male | 28(54.9) | 19(61.3) | 8(42.1) | 18(51.4) | 9(42.9) | 8(50.0) |  |
| Female | 23(45.1) | 12(38.7) | 11(57.9) | 17(48.6) | 12(57.1) | 8(50.0) |  |
| Age group(years) |  |  |  |  |  |  | 0.247 |
| <30 | 5(9.8) | 3(9.7) | 1(5.3) | 1(2.9) | 0(0) | 2(12.5) |  |
| 30–39 | 3(5.9) | 3(9.7) | 0(0) | 0(0) | 1(4.8) | 1(6.3) |  |
| 40–59 | 12(23.5) | 1(3.2) | 2(10.5) | 5(14.3) | 3(14.3) | 2(12.5) |  |
| 60–79 | 20(39.2) | 15(48.4) | 8(42.1) | 24(68.6) | 12(57.1) | 9(56.3) |  |
| ≥80 | 11(21.6) | 9(29.0) | 8(42.1) | 5(14.3) | 5(23.8) | 2(12.5) |  |
| At least one comorbidity | 25(49.0)[a] | 23(74.2) | 13(68.4) | 26(74.3) | 18(85.7) [a] | 11(68.8) | **0.034** |
| Temperature ≥39˚C | 30(58.8) | 16(51.6) | 6(31.6) | 16(45.7) | 9(42.9) | 8(50.0) | 0.444 |
| Cough | 51(100) | 31(100) | 19(100) | 34(97.1) | 21(100) | 16(100) | 0.705 |
| Sputum production | 39(76.5) | 29(93.5) | 15(78.9) | 30(85.7) | 19(90.5) | 16(100) | 0.120 |
| Pharyngalgia | 3(5.9) | 3(9.7) | 2(10.5) | 2(5.7) | 2(9.5) | 2(12.5) | 0.876 |
| Thoracalgia | 4(7.8) | 2(6.5) | 1(5.3) | 0(0) | 2(9.5) | 1(6.3) | 0.523 |
| Dyspnea | 0(0) [b] | 1(3.2) | 4(21.1) [b] | 1(2.9) | 1(4.8) | 0(0) | **0.007** |
| Runny nose | 1(2.0) | 1(3.2) | 1(5.3) | 1(2.9) | 0(0) | 2(12.5) | 0.360 |
| Vomiting | 0(0) | 3(9.7) | 0(0) | 3(8.6) | 1(4.8) | 1(6.3) | 0.123 |
| Presence of radiographic diagnosis of pneumonia | 38(74.5)[c] | 17(54.8) | 13(68.4) | 15(42.9) [c] | 13(61.9) | 7(43.8) | **0.042** |
| Visited a live poultry market | 1(2.0) | 1(3.2) | 0(0) | 0(0) | 0(0) | 0(0) | 0.880 |
| Contact with live poultry | 6(11.8) | 3(9.7) | 2(10.5) | 1(2.9) | 1(4.8) | 1(6.3) | 0.753 |
| Contact with a patient with fever | 3(5.9) | 4(12.9) | 2(10.5) | 1(2.9) | 3(14.3) | 2(12.5) | 0.442 |
| Current Smoker | 2(3.9) | 4(12.9) | 2(10.5) | 6(17.1) | 3(14.3) | 3(18.8) | 0.333 |
| Former Smoker | 10(19.6) | 7(22.6) | 2(10.5) | 7(20.0) | 3(14.3) | 0(0) |  |
| Never Smoked | 39(76.5) | 20(64.5) | 15(78.9) | 22(62.9) | 15(71.4) | 13(81.3) |  |

[*]*P* values denote comparisons among the six main pathogens. [a,b] and [c] signify *P*<0.05 for pairwise comparisons.

[a] refers to comparisons between the single-infected SARI patients with *M. pneumoniae* and those with Flu B/Yamagata.

[b] refers to comparisons between the single-infected SARI patients with *M. pneumoniae* and those with HRhV.

[c] refers to comparisons between SARI patients infected with *M. pneumoniae* and those infected with Flu A/H3N2.

## Seasonal trends

Fig 1 shows monthly variations in the number of SARI patients infected with *M. pneumoniae*, AdV, Flu A/H3N2, Flu A /pH1N1, HRhV, and Flu B/Yamagata. Over the 12-month period, the temporal distribution of pathogen-confirmed SARI patients had a bimodal shape, with the first peak in the summer and the second peak in the winter. The duration of the first positive peak was 2 months, from August to September, but the second peak lasted only 1 month. The infection peaks seemed to be attributed to the number of *M. pneumoniae* and AdV cases detected. In addition, Flu A/H3N2 contributed to the summer peak, whereas Flu B/Yamagata and Flu A/pH1N1 dominantly contributed to the winter peak. Unlike other pathogens, HRhV was detected all year along and did not show seasonal variations. The distributions of the seasonal patterns of the positivity rates of the main 6 pathogens are shown in Fig 2. Flu A/H3N2 prevalence peaked in the summer (Jun-Aug) and autumn (Sep-Nov), with positivity rates of 21.1% (20/95) and 22.3% (21/94), respectively (*P*>0.05). However, Flu A/pH1N1 and Flu B/

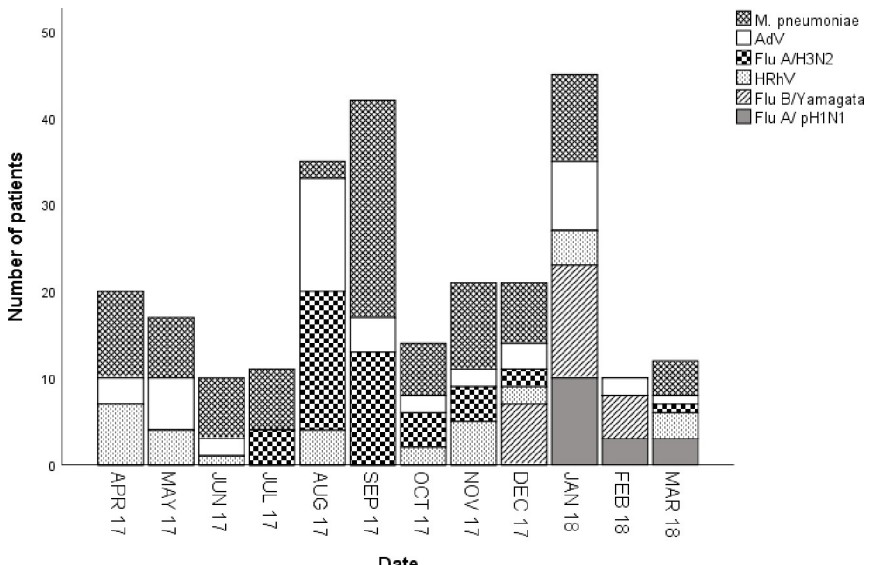

**Fig 1. Monthly variations of the six main pathogens detected in adult SARI patients in a surveillance hospital in Jinshan, Shanghai, April 2017 to March 2018.**

Yamagata peaked in the winter (Dec-Feb), with positivity rates of 9.8% (13/132) and 18.9% (25/132), respectively, the differences were statistically significant ($P<0.01$). It is worth noting that no SARI patients infected by Flu B/Yamagata were detected in the spring (Mar-May), summer or autumn. The positivity rate of *M. pneumoniae* was significantly higher in the autumn (43.6%, 41/94) than that in the other seasons ($P<0.01$). The positivity rate (18.4%, 14/76) of HRhV was significantly higher in spring than that in the other seasons ($P<0.01$). The positivity rate of AdV did not demonstrate obvious seasonality throughout the year ($P>0.05$).

## Age distribution

The age group distributions of the positivity rates of the main pathogens, *M. pneumoniae*, AdV, Flu A/H3N2, Flu A/pH1N1, HRhV, and Flu B/Yamagata, are shown in Fig 3. The prevalence rates of Flu A/pH1N1 (8.0%) and AdV (20.0%) peaked in the group younger than 30 years old, although the difference was not significant ($P>0.05$). The positivity rates of *M. pneumoniae* (36.2%) and Flu B/Yamagata (6.9%) were the highest in the 40-59-year-old group, without statistical significance ($P>0.05$). Moreover, no significant differences among the different age groups were observed with regard to the positivity rates of Flu A/H3N2 and HRhV. Interestingly, no patients infected with Flu A/H3N2 and HRhV was detected in the 30- to 39-year-old group.

## Treatment and prognosis

The median duration from illness onset to admission in SARI patients was 3 days (IQR: 2–5.5; range: 0 to 14 days), and the median duration of hospitalization was 10 days (IQR: 8–13 days). Complications occurred in 61 SARI patients, with electrolyte metabolism disorder (19 cases), respiratory failure (14 cases) and cardiac insufficiency (8 cases) being the most common complications. The remaining 336 patients did not report any complications. No significant differences between SARI patients with confirmed pathogens and those without confirmed pathogens was observed with regard to the use of antibiotics (levofloxacin, cephalosporin, azithromycin), antivirals (oseltamivir), glucocorticoids and oxygen therapy ($P>0.05$). The

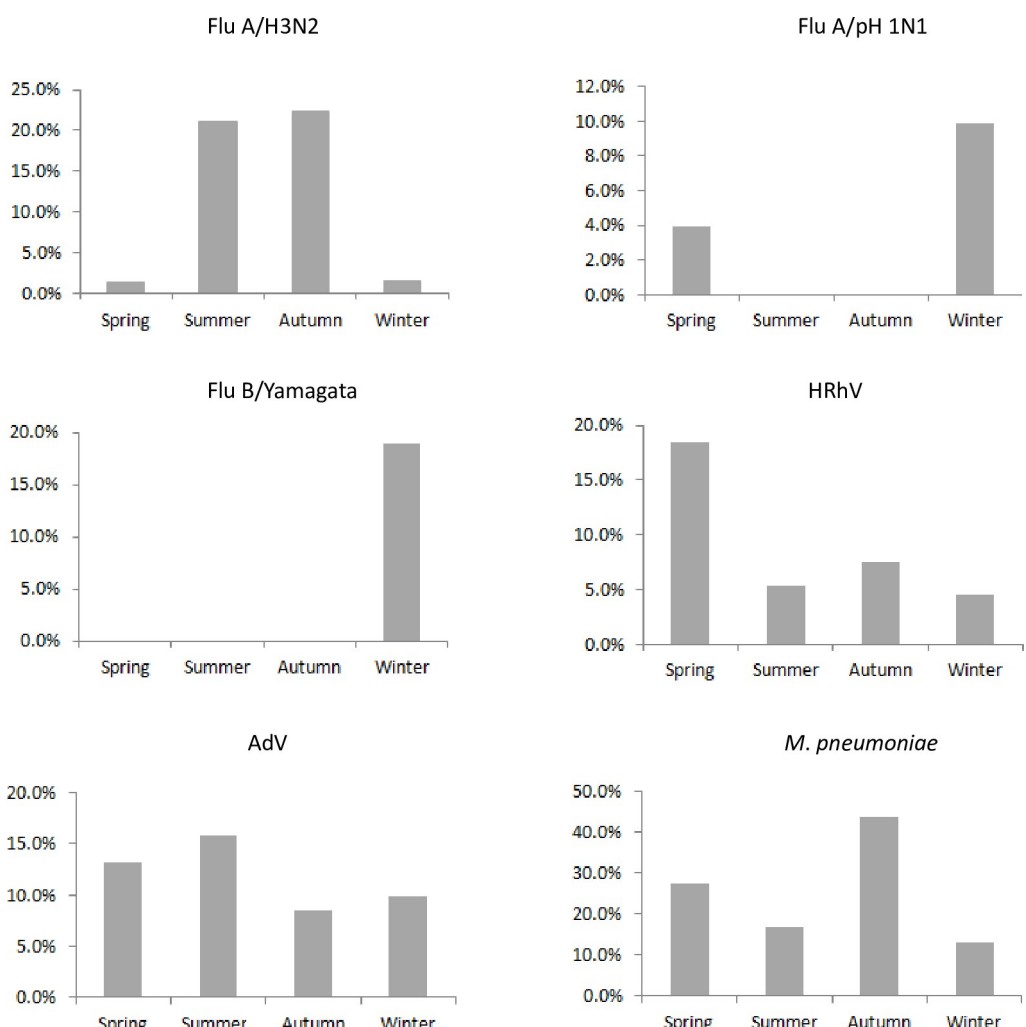

**Fig 2. Detection rates of the six main pathogens in adult SARI patients in different seasons in a surveillance hospital in Jinshan, Shanghai.** Each panel shows the seasonal distribution of a pathogen in SARI patients. For each pathogen, the detection rate on the y-axis refers to the number of positive patients divided by the total number of patients tested in a season.

duration of antibiotic use during hospitalization was 1–15 days (median: 9 days [IQR 5–11]) in SARI patients without confirmed pathogens and 1–20 days (median: 9 days [IQR 6–11]) in those with confirmed pathogens, though the difference was nonsignificant ($P = 0.68$). Three SARI patients died during hospitalization (Table 5).

## Discussion

Hospital-based sentinel surveillance of SARI can be used as a strategy to monitor trends in this relatively severe disease and is critical for establishing a platform to understand the epidemiological and etiological profiles at the local level. A monitoring study involving SARI patients in Georgia demonstrated that the proportions of patients positive for respiratory pathogens varied widely among seasons; there was no influenza detected in summer and early autumn (from July to October) but a 30% RSV positivity rate from March 2015–2017 [1]. Another surveillance study involving SARI patients in several countries found that the positivity rates of

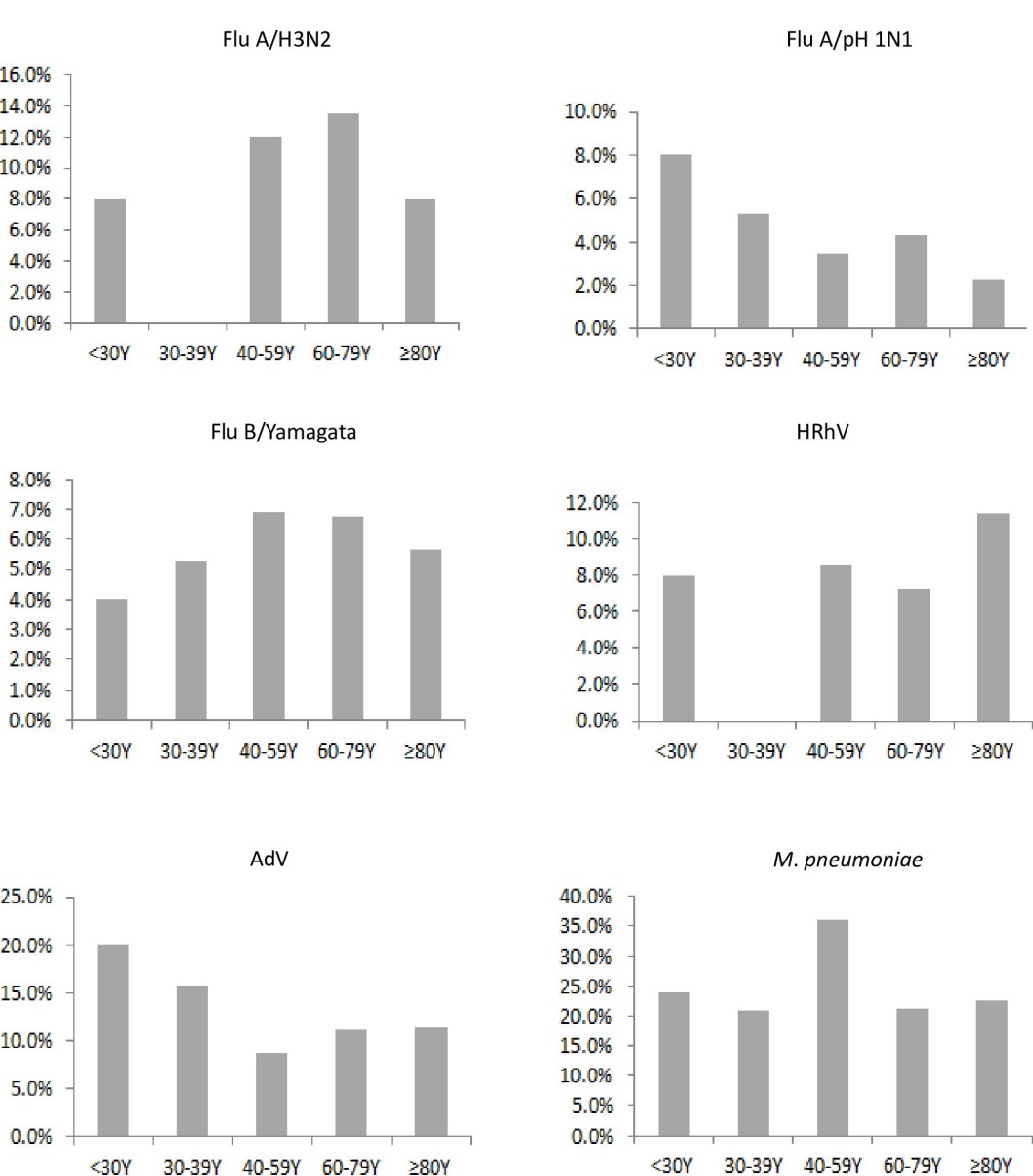

**Fig 3. Detection rates of the six main pathogens in SARI patients according to age group in a surveillance hospital in Jinshan, Shanghai, April 2017 to March 2018.** Each panel shows the age group-specific detection rate of one pathogen in SARI patients. For each pathogen, the detection rate on the y-axis refers to the number of positive patients divided by the total number of patients tested in each age group.

influenza viruses varied widely depending on country and season, from 2.1% in Armenia in 2011–2012 to 100% in Albania in 2009–2010 [18]. A comparative study of viral profiles in hospitalized pediatric SARI patients in Beijing and Shanghai, China, showed different viral profile patterns in the 2 cities; RSV (52.9%) and HRhV/enterovirus (34.7%) were the most prevalent etiological agents of SARI in Beijing, whereas HRhV/enterovirus (33.6%) and HBoV (17.7%) were the main pathogens of SARI in Shanghai [10]. The early detection of divergent SARI pathogens through sentinel surveillance can measure the burden of disease on the basis of severity and the better prepared region for an emergency response. To our knowledge, this is the first pilot study to continuously surveil 19 respiratory pathogens in adult SARI patients in

**Table 5. Treatments and prognoses in adult SARI patients in a surveillance hospital in Jinshan, Shanghai, April 2017 to March 2018.**

| Characteristics | SARI patients | | | P value* |
|---|---|---|---|---|
| | All (%) [n = 397] | With confirmed pathogens (%) [n = 250] | Without confirmed pathogens (%) [n = 147] | |
| Clinical course (median, days) | | | | |
| From illness onset to admission | 3 | 3 | 3 | 0.567 |
| Length of hospitalization | 10 | 10 | 10 | 0.545 |
| Antibiotics prior to hospitalization | 241(61.0) | 151(60.9) | 90(61.2) | 0.723 |
| Antibiotics during hospitalization | 393(99.0) | 246(98.4) | 147(100) | 0.301 |
| Antivirals | 11(2.8) | 7(2.8) | 4(2.7) | 1.000 |
| Glucocorticoids | 112(28.2) | 72(27.2) | 40(28.8) | 0.734 |
| Oxygen therapy | 196(49.4) | 124(49.6) | 72(49.0) | 0.918 |
| Complications | 61(15.4) | 37(14.8) | 24(16.3) | 0.684 |
| Death | 3(0.8) | 2(0.8) | 1(0.7) | 1.000 |

*P values denote comparisons between SARI patients with confirmed pathogens and SARI patients without confirmed pathogens.

Shanghai, eastern China, providing an improved understanding of the epidemiology, etiological spectrum and clinical profile of SARI. During 1 year of active surveillance, 397 patients who met the established case definition of SARI were eligible for enrollment in this study, and 63.0% of these patients tested positive for at least one pathogen. Our findings were in accordance with those reported elsewhere, which revealed etiologies in 50% to 85% of hospitalized SARI cases [7, 19, 20].

From April 2017 to March 2018 in Jinshan district, the main etiologies of SARI varied seasonally; *M. pneumoniae*, AdV, Flu A/H3N2, HRhV, Flu B/Yamagata, and Flu A/pH1N1 were the predominant pathogens depending on the month. Other viruses, such as PIV 1, HCoV NL63, HCoV 229E, HCoV HKU1, PIV 3, HMPV, PIV 4, HCoV OC43, Flu B/Victoria, RSV B and HBoV, were also present, although the numbers of patients infected with these viruses were relatively small. Since our surveillance system aimed to detect SARI in adult patients, most of the enrolled patients were elderly individuals aged 60–79 years (52.1%) 80 years and above (22.2%). Our study demonstrates that individuals over 60 years age are the most vulnerable to SARI in Jinshan, a subtropical region. In the present study, at least one chronic medical condition was present in 70% of SARI patients. Our study population had a high prevalence of comorbidities in comparison with those examined in Hubei Province study, China [12]. This may be partially explained by the socioeconomic development difference between the 2 regions. Hypertension and cardiovascular disease were observed in 38.3% and 7.6% of our population, respectively. Patients with confirmed pathogens had a higher prevalence of cardiovascular disease than those without confirmed pathogens. One study suggested that diagnosed cardiovascular disease was related to a fatal outcome in influenza-positive SARI patients [21]. Our study revealed that the proportions of patients who received influenza and pneumococcal conjugate vaccines were few, so respiratory disease vaccination programs targeting individuals with cardiovascular-related diseases should be recommended. In this study, most patients presented with cough, sputum production and fever. These clinical features bear some resemblance to those reported in a previous study [1]. It should be noted that empirical administration of antibiotics during hospitalization occurred in 99% of patients in the present study due to the unavailability of rapid pathogen identification tests. The current study found that pneumonia was the main reason for hospital admission of SARI patients (55.9%), followed by bronchiolitis (17.1%) in Jinshan, a region in eastern China. A similar study in northern China showed that pneumonia (88.95%) and bronchiolitis (6.37%) were also the top 2

admission features among SARI patients [22]. HRhV has emerged as an independent causative agent of lower respiratory tract infection. To date, the majority of investigations on HRhV-associated lower respiratory tract infection in adults have focused on immunocompromised patients [23–25] or those with hospital-acquired pneumonia [26, 27]. We compared the patients with single infection in terms of signs and symptoms, and the results showed that dyspnea was the most frequent symptom (21.1%) in community-acquired SARI patients infected by HRhV, which was consistent with the results of a similar multicenter study (30%) in China [28]. *M. pneumoniae* is an important cause of community-acquired pneumonia. Depending on the setting, 10–40% of community-acquired pneumonia patients are infected with *M. pneumoniae* [20]. Our study also showed that patients infected by *M. pneumoniae* had the highest rate of radiographic evidence of pneumonia (74.5%) compared with those infected by other single pathogens, demonstrating that community-acquired pneumonia is a heterogeneous disease. Among the 382 SARI patients who underwent chest CT, there was a significant difference in the proportion of patients who accepted a chest radiographic examination between SARI patients with confirmed pathogens and those without confirmed pathogens. However, a significant difference in the proportion of patients presenting a radiographic diagnosis of pneumonia between SARI patients with confirmed pathogens and those without confirmed pathogens was not observed, suggesting that the etiologies and disease courses of community-acquired pneumonia were highly variable.

*M. pneumoniae* (23.9%) was the most common pathogen in the present study. The positive detection rate of *M. pneumoniae* was similar to the published rate (19.7%) in northern China [20]. A prospective study in Hong Kong including adults hospitalized with pneumonia from 2004 to 2005 found that *M. pneumoniae* was detected in 78/1,193 patients (6.5%) [29]. *M. pneumoniae* occurs endemically worldwide in many different geographic regions. *M. pneumoniae* was mostly detected in autumn (43.6%) and spring (27.6%) in our study, but *M. pneumoniae* in Istanbul was more commonly identified in summer (44.9%) and winter (22.4%) [30]. As the second most common pathogen in this study, the positivity rate of AdV did not significantly differ seasonally; this trend in seasonality was consistent with previously reported AdV seasonality data from China [10]. In contrast with the seasonality of viral SARI observed in Georgia in 2015–2017 and in northern China in 2014–2016, where a distinct winter-only influenza peak was observed [31, 32], we found that influenza peaked in both the winter and in summer. Overall, influenza virus was common in this study, with Flu A/H3N2 dominating in summer and Flu B/Yamagata and Flu A/pH1N1 dominating in winter. According to our findings, the positivity rate of Flu B/Yamagata (18.9%) was nearly twice that of Flu A/pH1N1 (9.8%) in winter; this result was different from that of a study in the USA in which estimated excess hospitalization rates associated with influenza B were lower than those associated with Flu A/H3N2 [33]. In this study, we also noted that no significant differences were found in the positivity rates of *M. pneumoniae*, AdV, Flu A/H3N2, Flu A /pH1N1, HRhV, and Flu B/Yamagata among the different age groups. This result was basically the same as that in a previous study in China [10] and may be attributed to susceptibility to these common viruses in different age groups of adults. As reported elsewhere [34], coinfections were relatively common in the present study. A total of 13.1% of SARI patients were reported to have more than one pathogen infection; this percentage was consistent with that in a previous study (11.7%) [19].

## Limitations

Our study was subject to several limitations. First, as a pilot study, this study was conducted at only 1 hospital. Even though this hospital is the largest hospital in the district, the findings may have relatively limited generalizability. The prevalence of each pathogen may vary in regions

with different climates, demographic patterns and accessibility to healthcare. Second, the result was based on SARI surveillance over a 12-month period, and the burden due to SARI may not reflect the actual situation over several years. Third, the case report form in this study was a standard structured questionnaire, and the results were collected to determine whether the patient had received a radiographic diagnosis of pneumonia. It was impossible to pinpoint the type of pneumonia, such as lobar pneumonia or atypical pneumonia. The pathogens detected in this pilot study covered only common respiratory viruses and *M. pneumoniae* and did not include related respiratory bacterial pathogens, such as *Pneumococcus* and *Bordetella pertussis*, owing to limited financial support, so SARI patients without confirmed pathogens may have been positive for other nontested bacterial pathogens. Indeed, the inclusion of bacterial surveillance is under consideration for integration into our program.

## Conclusions

In conclusion, the current study was the first to monitor hospitalized adult SARI patients for most respiratory viruses and *M. pneumoniae* in Shanghai and confirmed that multiple respiratory pathogens may circulate among the SARI population and vary with climatic and demographic characteristics. This finding highlights the importance of sustained sentinel surveillance of SARIs at the local and national levels, which can guide accurate evaluations of the prevalence of etiological agents of SARI and the burden of disease and, most importantly, shape public policies on SARI prevention and control.

## Supporting information

**S1 Dataset. Minimal data set.**
(XLSX)

**S1 File. Sequences of primers targeting Flu A/B used in real-time RT-PCR.**
(DOC)

**S2 File. STROBE statement—checklist of items that should be included in reports of observational studies.**
(DOCX)

## Acknowledgments

We would like to express our gratitude to the physicians and nurses from Jinshan District Central Hospital for their significant contribution to the data collection and specimen collection.

## Author Contributions

**Conceptualization:** Jian Li, Jian-Min Zhu.

**Data curation:** Tang Wang, Shu-Hua Li.

**Formal analysis:** Jian Li, Can-Lei Song, Tang Wang.

**Funding acquisition:** Can-Lei Song, Shu-Hua Li.

**Investigation:** Can-Lei Song, Tang Wang, Shu-Hua Li.

**Methodology:** Yu-Long Ye, Jian-Ru Du.

**Writing – original draft:** Jian Li, Can-Lei Song, Tang Wang.

**Writing – review & editing:** Jian Li, Jian-Min Zhu.

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
