## [Decision Letter · Decision Letter 0]

24 Nov 2020

PONE-D-20-19561

Etiological and epidemiological characters of severe acute respiratory infection caused by multiple viruses and mycoplasma pneumoniae in adult patients in Jinshan of Shanghai, April 2017 to March 2018: a pilot hospital-based surveillance study

PLOS ONE

Dear Dr. LI,

Thank you for submitting your manuscript to PLOS ONE. After careful consideration, we feel that it has merit but does not fully meet PLOS ONE’s publication criteria as it currently stands. Therefore, we invite you to submit a revised version of the manuscript that addresses the points raised during the review process.

I have received the reviews of your manuscript. While your paper addresses an interesting question, the reviewers stated several concerns about your study and did not recommend publication in its present form. There are concerns regarding the used methodology for data analysis as well as data presentation, and these comments need to be addressed carefully. Please see reviewers’ insightful comments below.  Additionally, I have several concerns regarding this study that also need to be addressed carefully.  In addition, the quality of the language needs to be improved since there are awkward sentences, typo and throughout the manuscript (see attached PDF file). Please have a fluent, preferably native, English-language speaker thoroughly copyedit your manuscript for language usage, spelling, and grammar.  

Specific comments:

Specimen collection and laboratory testing: This section need further clarification. Please specify the multiplex PCR used. Did the authors used method described in previous literatures or commercial kit?Line 219 – 223: Not sure what the authors wish to convey, please rephrase for clarification.Line 260 – 265:  Not clear on what the authors’ intention on these statement, please clarify.Line 295 – 299: This argument does not hold.  These are not fair comparison since this study excluded children.Ethical statement:  This needs to be included in the Materials and Methods section and needs to include approval number.

We look forward to receiving your revised manuscript.

Kind regards,

Baochuan Lin, Ph.D.

Academic Editor

PLOS ONE

Journal Requirements:

2. In your methods and ethics statement, please state whether you obtained consent from parents or guardians of minors under 18 years old.

3. PLOS ONE requires experimental methods to be described in enough detail to allow suitably skilled investigators to fully replicate and evaluate your study. See https://journals.plos.org/plosone/s/submission-guidelines#loc-materials-and-methods for more information.

To comply with PLOS ONE submission guidelines, in your Methods section, please provide a more detailed description of your methodology, specifically about your respiratory pathogens 15 multiplex real-time RT-PCR, Flu A/B RT-PCR, and flu typing methods.

Reviewers' comments:

Reviewer's Responses to Questions

**Comments to the Author**

1. Is the manuscript technically sound, and do the data support the conclusions?

Reviewer #1: Yes

Reviewer #2: Yes

2. Has the statistical analysis been performed appropriately and rigorously? 

Reviewer #1: Yes

Reviewer #2: Yes

3. Have the authors made all data underlying the findings in their manuscript fully available?

Reviewer #1: Yes

Reviewer #2: Yes

4. Is the manuscript presented in an intelligible fashion and written in standard English?

Reviewer #1: Yes

Reviewer #2: Yes

5. Review Comments to the Author

Reviewer #1: Dear Author

Thank you for the very nice work, indeed it generated comprehensive and very informative data. The active surveillance is much appreciated. I understand that such surveillance produced a lot of data which I believe is a big challenge to make the best out of it which you did through a very nice data presentation and analysis. In addition SARI surveillance in adult is not addressed much in the literature especially in developing areas. Moreover it seems that you described surveillance from a special geographical area characterized with unique pattern of SARI surveillance especially for the influenza B as well as the summer seasonal influenza H3 peak.

Comments:

1- The 1st letters in the title are to be capitalized.

Abstract

1- In the abstract line 71-73, the statement “No significant difference among … rate of main

pathogens.” is unclear, please rephrase.

Methods

2- Line 132 please insert a reference for Sari definition.

3- Please specify details of sample collection: oropharyngeal or nasopharyngeal or both, type of the swabs used and manufacturer, VTM inhouse prepared or commercial and it’s manufacturer, duration of sample storage till transportation.

4- Please specify the type of kits used : catalogue number, manufacturer or if it is inhouse made, provide primers and reagent used along with the reference.

5- Study subjects: Are the patients admitted in ICU or regular wards?

6- Line 158-159 “Specimens were lysed at strongly denaturing conditions to deactivate RNases” please provide a reference as I believe that harsh conditions may affect the target fragile viral RNA.

7- Line 160: using term “contaminant” is incorrect

Results

8- Line 237: it is not clear where did these numbers came from (20/95, 21/94) and how can the P value show significant difference between these very close findings. Please recheck and clarify.

9- Line 239 and 240 please clarify what this P value indicates.

Discussion

10- For the significant P values, you addressed the comorbidities in the discussion. What about the dyspnea and the radiologic examination.

11- Findings in the result section line 224 and 225 were not discussed regarding the Xray finding in the mycoplasma and rhino causing dyspnea.

12- In the discussion, comparison of the patients from Madagascar and yours is irrelevant as they enrolled pediatric patients that were excluded from your study.

13- Line 311: You discuss cough as being the most common symptom, this is obvious as it in part of the inclusion criteria. Rather, you should address elaboration about the pneumonia and bronchiolitis.

Figures and tables:

14- Figure 3: Percentage of the y axis is not clear (is it from the total enrolled or from the positive cases only). Please provide your definition of the detection rate.

15- Table 5: please draw lines between columns as it is confusing.

16- Table 4: Title is not informative. Significant P values need further analysis to detect the significance is between which 2 groups.

17- Table3: It is not clear what is meant by “Chest radiographic exam”, please clarify especially that it shows significant P value and should be addressed in the discussion.

18- In table 2 : Percent is done from the total enrolled cases or from the positive ones. Please clarify and add the total number at the end.

GENERAL:

19- Please specify that the surveillance addresses the community acquired infections.

20- When you mention “Presence of radiographic diagnosis of pneumonia” you mean, lobar pneumonia denoting mostly bacterial origin , or atypical pneumonia denoting viral or atypical bacterial origin (Mycoplasma). These details need to be mentioned especially for the negative cases as they may indicate other non-tested bacterial etiology.

21- Some sentences are ambiguous and need to be rephrased or corrected:

a. Line 149

b. Line 188: remove “positive”

c. Line 191

d. Line 273-274

e. Line 295

f. Line 323

g. Line 341

h. Line 345

Recommendations:

- The title include many details that can be removed as the age group and the study period

- Seasonality is better described in Epidemiologic weeks (Epi-weeks)

Reviewer #2: The authors described the etiological and epidemiological characters of severe acute respiratory infection caused by multiple viruses and mycoplasma pneumoniae in adult patients in Jinshan of Shanghai, April 2017 to March 2018. So befor publication there are some points need to revise as following:-

Major questions Must be clarified:-

1- Why did the authors not represent the values of real time PCR / RT-PCR for the detected pathogens as an indicator for the load of different pathogens  and if there are variations among their load in relation to seasonal variation?

2- Only pathogens from males (173 positive cases) were statistically analyzed in relation to different variants such as type of pathogen, clinical and diagnostic parameters, age......etac Why did not authors do the same data analysis for female samples (77 positive cases) as in table 4? Also, Table 1 based manily on male cases (194) and no data concerning the female (203), why?

3- Among 19 pathogens have been detected authors decided to focus on only 6 pathogens although other studies stated the predominance of other pathogens such as RSV?

Minor comments

The manuscript should be revised carfeulley for typographical errorrs.

Abstract

abbravietions in line 71 should be defined at its first apperance as in lline 66 then use the abbrevations

lines 66-67 only 217 pathogens reported while in line 63 they are 250, could you mention the other typeof etological agent and its frequency.

Background

line 100:- "owing to the lack of gold standard methods to swiftly determine etiological diagnoses" change to "owing to the lack of gold standard diagnostic methods to swiftly determine etiological agents"

Materials and methods

Line 133:- "≥38˚C, cough, with onset within the last 10 days and require hospitalization" change to "≥38˚C, cough onset within the last 10 days and require hospitalization"

Lines 137-138:- "vaccination (vaccinating influenza vaccine during 1 year before illness onset, vaccinating pneumococcal conjugate vaccine)" change to "vaccination (receiving influenza vaccine during 1 year before illness onset,and pneumococcal conjugate vaccine)"

Line 149:- "149 information that could identify the identification of patients was masked during or after data" change to "149 information that could identify the personality of patients was masked during or after data"

Line 157:- "viral RNA and DNA using the QIAamp Viral RNA Mini Kit (Qiagen, Hilden, Germany) following " change to "viral RNA and DNA using the QIAamp Viral RNA/DNA Mini Kit (Qiagen, Hilden, Germany) following"

Lines 161-162:- "Total nucleic acid extracts were further processed by multiplex real-time reverse transcription" change to "Viral nucleic acid extracts were further processed by multiplex real-time reverse transcription" since you used kit for viral nucleic acid (RNA or DNA)

Lines 163-163:- "Respiratory pathogens 15 multiplex real-time RT-PCR diagnostic strategy was adopted to detect PIV (types 1, ......." change to "The multiplex real-time RT-PCR diagnostic strategy was adopted to detect 15 respiratory pathogens, PIV (types 1, ......."

Results

As general when you describe the results please make full description of the full cases either positive or not and do not leave unclear such as line 212 you mentioned 382 cases and ignored the residue 15 cases and this was repeated allover the manuiscript, do not leave anything for guessing.

Lines 199-203:- Authors described the frequency and type of pathogens,however in compare to table 2 there is confusion concerning the pathogen frequency as in text 198 singl and 52 multiple, while later on the number will be 232 and in table 312, how can this occur? please clarify this.

lines 213-215:- "Thirty-two SARI patients and 30 patients had exposure of contacting with patients with fever and respiratory symptoms and contacting with live poultry during 2 weeks before their illness onset, respectively" change to "Thirty-two SARI patients had exposures with fever and respiratory symptoms patients while 30 SARI patients contacted with live poultry during 2 weeks before their illness onset"

Tables

1- Table 1 1st row change " All SARI SARI patient with confirmed pathogens SARI patient without confirmed pathogens" to "All with confirmed pathogens without confirmed pathogens" and add SARI patient above as another row.

2- Table 2 1st clonumn please change "viral etiology" to "etiology" only because there is a bacteria also mentioned there.

3- Table 3 1st row change " All SARI SARI patient with confirmed pathogens SARI patient without confirmed pathogens" to "All with confirmed pathogens without confirmed pathogens" and add SARI patient above as another row. Visiting a live poultry market and Contact with live poultry in table 3 looks the same where in table 4 it become one catogery Contact with live poultry.

Figures

The presented pathogens in Fig 1-3 based only on male SARI cases with confirmed pathogens or included all pathogens from male and female cases.

6. PLOS authors have the option to publish the peer review history of their article (what does this mean?). If published, this will include your full peer review and any attached files.

Reviewer #1: No

Reviewer #2: No

---

## [Author Response · Author response to Decision Letter 0]

17 Jan 2021

Response to Specific Comments:

1. Specimen collection and laboratory testing: This section need further clarification. Please specify the multiplex PCR used. Did the authors used method described in previous literatures or commercial kit?

Answer: We thank for these suggestions and have made further clarification. The multiplex PCR used is the commercial kit. We added the data about multiplex PCR and made further clarification (see Page 7, line 154 to Page 8, line 156 in Revised Manuscript with Track Changes, the same below). 

Line 219 – 223: Not sure what the authors wish to convey, please rephrase for clarification.

Answer: We are sorry and have rephrased these sentences (see Page 11, line232-236).

2. Line 260 – 265: Not clear on what the authors’ intention on these statement, please clarify.

Answer: These sentences in line 260-265 mean to show that there were no significant differences of therapy between SARI patients with confirmed pathogen and those without confirmed pathogen. We have modified our text as advised (see Page 13, line 280-286).

3. Line 295 – 299: This argument does not hold. These are not fair comparison since this study excluded children.

Answer: This comment is appreciated highly. We deleted these sentences in line 295-298 following this comment, and revised the next sentence in line 298-299(see Page 15, line321-323).

4. Ethical statement: This needs to be included in the Materials and Methods section and needs to include approval number.

Answer: The ethical statement has been moved to the Materials and Methods section, and the approval number has been added (see Page 9, line 181-186).

Response to Journal Requirements: 

 Answer: We ensure that our manuscript meets the journal’s style.

2. In your methods and ethics statement, please state whether you obtained consent from parents or guardians of minors under 18 years old.

 Answer: We have stated that the consent from parents or guardians of those under 18 years old have been obtained in the section of “ethics statement” (see Page 9, line 184-186).

3. PLOS ONE requires experimental methods to be described in enough detail to allow suitably skilled investigators to fully replicate and evaluate your study. See https://journals.plos.org/plosone/s/submission-guidelines#loc-materials-and-methods for more information.

To comply with PLOS ONE submission guidelines, in your Methods section, please provide a more detailed description of your methodology, specifically about your respiratory pathogens 15 multiplex real-time RT-PCR, Flu A/B RT-PCR, and flu typing methods. 

 Answer: We have provided a more detailed description of methodology in the section of specimen collection and laboratory testing as advised (see Page 7, line 142 to Page 8, line 166). 

 Answer: We agree to provide the minimal anonymized data set as Supporting Information files for data-sharing. And Data Availability statement has been updated, and you can revise it on our behalf.

 Answer: We have moved the ethics statement to the Methods sections of manuscript.

 Answer: We have added captions for the Supporting Information files at the end of the revised manuscript (see Page 20, line 420-421), and updated in-text citations as advised. 

Response to Reviewer #1' comments: 

Reviewer #1: Dear Author

Thank you for the very nice work, indeed it generated comprehensive and very informative data. The active surveillance is much appreciated. I understand that such surveillance produced a lot of data which I believe is a big challenge to make the best out of it which you did through a very nice data presentation and analysis. In addition SARI surveillance in adult is not addressed much in the literature especially in developing areas. Moreover it seems that you described surveillance from a special geographical area characterized with unique pattern of SARI surveillance especially for the influenza B as well as the summer seasonal influenza H3 peak.

Comments:

1- The 1st letters in the title are to be capitalized.

Answer: The first letters in the title have been capitalized as advised.

Abstract

1- In the abstract line 71-73, the statement “No significant difference among … rate of main pathogens.” is unclear, please rephrase.

Answer: We have modified the statement of this sentence (see Page 3, line 62-64).

Methods

2- Line 132 please insert a reference for Sari definition.

Answer: We thank for this suggestion. A reference for SARI definition has been inserted (see Page 6, line 123).

3- Please specify details of sample collection: oropharyngeal or nasopharyngeal or both, type of the swabs used and manufacturer, VTM inhouse prepared or commercial and it’s manufacturer, duration of sample storage till transportation. 

Answer: We have specified the details of sample collection including the type of swab and manufacturer. The information of VTM manufacturer and duration of sample storage till transportation have been provided as advised (see Page 7, line 142-147). 

4- Please specify the type of kits used : catalogue number, manufacturer or if it is inhouse made, provide primers and reagent used along with the reference.

Answer: The information of PCR kits has been specified (see Page 7, line 154 to Page 8, line 156). Both of the primers and reagent came from the PCR kit. The testing process of PCR was conducted according to the manufacturer’s protocols.

5- Study subjects: Are the patients admitted in ICU or regular wards?

Answer: The patients in this study included those admitted in ICU, respiratory medicine department and general wards, which was specified in the Study Subject section (see Page 6, line 118-119). 

6- Line 158-159 “Specimens were lysed at strongly denaturing conditions to deactivate RNases” please provide a reference as I believe that harsh conditions may affect the target fragile viral RNA.

Answer: We have followed the comment, deleted the term of “strongly” and rephrased the sentence in line 158, also, a reference has been provided according to your suggestion (see Page 7, line 151-152).

7- Line 160: using term “contaminant” is incorrect

Answer: Another reviewer thought that it was unnecessary to keep the sentence which was located in line 159-160, namely, “After adding alcohol and loading lysates onto the QIAamp spin column, viral RNA and DNA combined to the QIAamp silica membrane while contaminants passed through”. We followed this suggestion and deleted this sentence which included the term of “contaminant”. 

Results

8- Line 237: it is not clear where did these numbers came from (20/95, 21/94) and how can the P value show significant difference between these very close findings. Please recheck and clarify.

Answer: The denominator (95,94) were the total number of monitoring patients in summer(Jun-Aug) and autumn(Sep-Nov) respectively, and the numerator(20,21) were the positive number of patients in summer(Jun-Aug) and autumn(Sep-Nov) respectively. As for the P value, we are sorry for negligence. The P value should be 0.83 and the difference is not significant. Thanks for point to this mistake, we have corrected it (see Page 12, line 254). 

9- Line 239 and 240 please clarify what this P value indicates.

Answer: We have clarified the significance of this P value (see Page 12, line 254-256).

Discussion

10- For the significant P values, you addressed the comorbidities in the discussion. What about the dyspnea and the radiologic examination.

Answer: We thanks for this comment. We have addressed the dyspnea and presence of radiographic diagnosis of pneumonia in the discussion (see Page 16, line 344 to Page 17, line 364).

11- Findings in the result section line 224 and 225 were not discussed regarding the Xray finding in the mycoplasma and rhino causing dyspnea.

Answer: We thanks for this comment and have discussed them accordingly (see Page 16, line 344 to Page 17, line 357).

12- In the discussion, comparison of the patients from Madagascar and yours is irrelevant as they enrolled pediatric patients that were excluded from your study.

Answer: This comment is appreciated and we deleted this comparison in the discussion.

13- Line 311: You discuss cough as being the most common symptom, this is obvious as it in part of the inclusion criteria. Rather, you should address elaboration about the pneumonia and bronchiolitis.

Answer: We are sorry for no discussing the pneumonia in discussion on account of space limitation of original manuscript. In the revised paper, we have discussed the pneumonia and bronchiolitis following the suggestion (see Page 16, line 340-344).

Figures and tables:

14- Figure 3: Percentage of the y axis is not clear (is it from the total enrolled or from the positive cases only). Please provide your definition of the detection rate.

Answer: We have clarified the significance of y axis and provided the definition of the detection rate in Fig 2 and Fig3.

15- Table 5: please draw lines between columns as it is confusing.

Answer: We have drawn lines between columns in all 5 tables according to this comment(see Table 5).

16- Table 4: Title is not informative. Significant P values need further analysis to detect the significance is between which 2 groups.

Answer: Title of table 4 has been revised (see Page 31, line 609-610). As for 3 variables with significant P value, we conducted the pairwise comparison (see Page 32, line 611-615). Also, we revised the statistics section accordingly (see Page 9, line 177-178).

17- Table3: It is not clear what is meant by “Chest radiographic exam”, please clarify especially that it shows significant P value and should be addressed in the discussion.

Answer: It means the acceptance of chest radiographic exam, we have revised it and clarified especially in bold font in table 3. And we addressed it in the discussion (see Page 17, line 357-364). 

18- In table 2 : Percent is done from the total enrolled cases or from the positive ones. Please clarify and add the total number at the end.

Answer: Percent refers to the frequency of positive etiology divided by the total enrolled samples (397 cases). We have provided the explanation for it under the table 2 and added the total number at the end (see Page 29, line 590-591).

GENERAL:

19- Please specify that the surveillance addresses the community acquired infections.

Answer: We have specified this important significance of surveillance system in the Background section (see Page 5, line 99-101).

20- When you mention “Presence of radiographic diagnosis of pneumonia” you mean, lobar pneumonia denoting mostly bacterial origin, or atypical pneumonia denoting viral or atypical bacterial origin (Mycoplasma). These details need to be mentioned especially for the negative cases as they may indicate other non-tested bacterial etiology.

Answer: We are sorry that our case report form is the standard structural questionnaire, and it just collected the result whether has the presence of radiographic diagnosis of pneumonia, and can not show lobar pneumonia or atypical pneumonia. Meanwhile, the pathogens tested in this piloting study only covered common respiratory viruses and Mycoplasma pneumonia, and did not include respiratory bacterium. We agreed this comment and we address it in the limitation section (see Page 19, line 400-408).

21- Some sentences are ambiguous and need to be rephrased or corrected:

a. Line 149

Answer: The sentence in line 149 has been revised (see Page 7, line 139-140).

b. Line 188: remove “positive”

Answer: The term of “positive” in line 188 has been removed (see Page 9, line 190).

c. Line 191

Answer: The sentence in line 191 has been revised (see Page 9, line 194-195).

d. Line 273-274

Answer: The sentence in line 273-274 has been revised (see Page 14, line 295-297).

e. Line 295

Answer: The previous comment thought the sentence in line 295 did not hold, so we delete this sentence in line295-298. 

f. Line 323

Answer: The “viral respiratory SARI” in line 323 has been changed to “viral SARI” (see Page 18, line 376).

g. Line 341

Answer: The sentence in line 341 has been revised (see Page 19, line 397-398).

h. Line 345

Answer: The sentence in line 345 has been revised (see Page 19, line 404-407).

Recommendations:

1- The title include many details that can be removed as the age group and the study period

Answer: We deleted the study period (April 2017 to March 2018) from the title following the recommendation. Meanwhile, we respect the editor’s suggestion about this point. Since SARI surveillance in adults is not addressed much in the literatures especially in developing areas, we think it’d better to keep ‘adult’ in the title to show the difference from other studies. 

2- Seasonality is better described in Epidemiologic weeks (Epi-weeks)

Answer: We respect this recommendation and it is accepted that seasonality can be described in both weeks and months. Some studies about SARI surveillance described seasonality in months, such as reference of 10 and 20. Also , our piloting study only last for 12 months and did not include enough patients. In the case of relatively small sample size of patients with confirmed pathogens, the use of weeks will make the seasonality character can not be better displayed. So we thought it is better to describe seasonality in months in order to show the characteristics of seasonality of SARI clearly. 

Response to Reviewer #2' comments: 

Reviewer #2: The authors described the etiological and epidemiological characters of severe acute respiratory infection caused by multiple viruses and mycoplasma pneumoniae in adult patients in Jinshan of Shanghai, April 2017 to March 2018. So befor publication there are some points need to revise as following:-

Major questions Must be clarified:-

1- Why did the authors not represent the values of real time PCR / RT-PCR for the detected pathogens as an indicator for the load of different pathogens and if there are variations among their load in relation to seasonal variation?

Answer: The PCR kit this study used is a qualitative detection kit. The detecting results were judged by Tm value of various pathogens according to melting curve. The kit didn’t provide the quantitative value for the load of different pathogens. So, we are sorry that we can’t state if there are variations among loads in relation of seasonal variation. We have clarified the qualitative characteristic of PCR kit in the manuscript following in this comment (see Page 7, line 154 to Page 8, line 156).

2- Only pathogens from males (173 positive cases) were statistically analyzed in relation to different variants such as type of pathogen, clinical and diagnostic parameters, age......etac Why did not authors do the same data analysis for female samples (77 positive cases) as in table 4? Also, Table 1 based manily on male cases (194) and no data concerning the female (203), why?

Answer: Please allow us to clarify these problems. Both of the differences between males and females for the proportions in table 4 and table 1 have been analyzed, and initially, we omitted to display the information of female patients on consideration of controlling the length of table. We have added a row to show the female information in table 1 and table 4. 

3- Among 19 pathogens have been detected authors decided to focus on only 6 pathogens although other studies stated the predominance of other pathogens such as RSV?

Answer: This study detected 17 kinds of pathogens, in which the number of six pathogens exceeds 10. So we focus on these 6 main pathogens as the number of other seven pathogens all was fewer than 10. Table 2 described all detected pathogens. We have clarified this in the discussion (see Page 15, line 313-319).

Minor comments

1-The manuscript should be revised carefulley for typographical errors.

Answer: We have revised carefully for typographical error of the manuscript.

Abstract

2-abbravietions in line 71 should be defined at its first appearance as in line 66 then use the abbreviations

Answer: The names of viruses in line 71 have been defined with their full names at their first appearance (see Page 2, line 38 and Page 3, line 49-52). Other abbreviations in the manuscript have also been checked and revised.

3-lines 66-67 only 217 pathogens reported while in line 63 they are 250, could you mention the other type of etological agent and its frequency.

Answer: 217 pathogens in lines 66-67 refers to the total frequency of 4 main pathogens, and 250 in line 63 is the total number of patients who were identified as at least 1 pathogen. We have followed this suggestion and added the other type of etiological agents and their frequency in the abstract (see Page 3, line 51-57).

Background

4-line 100:- "owing to the lack of gold standard methods to swiftly determine etiological diagnoses" change to "owing to the lack of gold standard diagnostic methods to swiftly determine etiological agents"

Answer: We thanks for this suggestion and revised this sentence accordingly(see Page 4, line 84-85). 

Materials and methods

5-Line 133:- "≥38˚C, cough, with onset within the last 10 days and require hospitalization" change to "≥38˚C, cough onset within the last 10 days and require hospitalization"

Answer: We thanks for this suggestion and revised this sentence accordingly (see Page 6, line 122-123).

6-Lines 137-138:- "vaccination (vaccinating influenza vaccine during 1 year before illness onset, vaccinating pneumococcal conjugate vaccine)" change to "vaccination (receiving influenza vaccine during 1 year before illness onset,and pneumococcal conjugate vaccine)"

Answer: We thanks for this suggestion and revised this sentence accordingly (see Page 6, line 127-128).

7-Line 149:- "149 information that could identify the identification of patients was masked during or after data" change to "149 information that could identify the personality of patients was masked during or after data"

Answer: We thanks for this suggestion and revised this sentence accordingly (see Page 7, line 139-140).

8-Line 157:- "viral RNA and DNA using the QIAamp Viral RNA Mini Kit (Qiagen, Hilden, Germany) following " change to "viral RNA and DNA using the QIAamp Viral RNA/DNA Mini Kit (Qiagen, Hilden, Germany) following"

Answer: We are sorry for this negligence and revised this sentence according to the suggestion (see Page 7, line 148-150).

9-Lines 161-162:- "Total nucleic acid extracts were further processed by multiplex real-time reverse transcription" change to "Viral nucleic acid extracts were further processed by multiplex real-time reverse transcription" since you used kit for viral nucleic acid (RNA or DNA)

Answer: We thanks for this suggestion and revised this sentence accordingly(see Page 7, line 153-154).

10-Lines 163-163:- "Respiratory pathogens 15 multiplex real-time RT-PCR diagnostic strategy was adopted to detect PIV (types 1, ......." change to "The multiplex real-time RT-PCR diagnostic strategy was adopted to detect 15 respiratory pathogens, PIV (types 1, ......."

Answer: We thanks for this suggestion and revised this sentence following the suggestion (see Page 7, line 154 to Page 8, line 156).

Results

11-As general when you describe the results please make full description of the full cases either positive or not and do not leave unclear such as line 212 you mentioned 382 cases and ignored the residue 15 cases and this was repeated allover the manuiscript, do not leave anything for guessing.

Answer: We thank for this suggestion, and have tried our best to clarify these unclear descriptions all over the manuscript as advised (see Page 9, line 191-197; Page 10, line 199-200; Page 11, line 222-225; Page 13, line 277-280 ). 

12-Lines 199-203:- Authors described the frequency and type of pathogens,however in compare to table 2 there is confusion concerning the pathogen frequency as in text 198 singl and 52 multiple, while later on the number will be 232 and in table 312, how can this occur? please clarify this.

Answer: Number of 198 and 52 in line 199 were the number of patients with single and multiple infections, respectively. Numbers from line 201 to line 203 including 95 (M. pneumoniae), 46 (AdV), 44 (Flu A/H3N2), 32 (HRhV), 25 (Flu B/Yamagata) represent the frequency of identified pathogen which was detected most frequently, and their meaning was different from that in line 199.Numbers from the 3rd row( 16 for Flu A/pH1N1) to the 25th row(95 for M. pneumoniae) in table 2 also represent the frequency of identified pathogens and their total number equals to 312. We have revised the corresponding description in section of etiology (see Page 10, line 206-214), and added the explanation for frequency under the table 2.

13-lines 213-215:- "Thirty-two SARI patients and 30 patients had exposure of contacting with patients with fever and respiratory symptoms and contacting with live poultry during 2 weeks before their illness onset, respectively" change to "Thirty-two SARI patients had exposures with fever and respiratory symptoms patients while 30 SARI patients contacted with live poultry during 2 weeks before their illness onset"

Answer: We thanks for this suggestion and revised this sentence following the suggestion (see Page 11, line 225-227).

Tables

1- Table 1 1st row change " All SARI SARI patient with confirmed pathogens SARI patient without confirmed pathogens" to "All with confirmed pathogens without confirmed pathogens" and add SARI patient above as another row.

Answer: We have revised the 1st row of Table 1 and added SARI patient above as another row following this suggestion (see Table1).

2- Table 2 1st clonumn please change "viral etiology" to "etiology" only because there is a bacteria also mentioned there.

Answer: We are sorry for this negligence and have changed it according to the suggestion (see Table 2).

3- Table 3 1st row change " All SARI SARI patient with confirmed pathogens SARI patient without confirmed pathogens" to "All with confirmed pathogens without confirmed pathogens" and add SARI patient above as another row. Visiting a live poultry market and Contact with live poultry in table 3 looks the same where in table 4 it become one catogery Contact with live poultry.

Answer: We have revised the 1st row of Table 3 and added SARI patient above as another row following this suggestion, so does the Table 5. Contact with live poultry included contacting with live poultry at home and other place (such as live poultry market), so it is different from visiting a live poultry market. Since the number of patients visiting a live poultry market was just 3 cases, and it only included 1 case with single-infected M. pneumoniae positivity and 1 case with single-infected AdV positivity, the third case belonged to multiple infections, so the initial table 4 didn’t analyze this variable. We have analyzed it in table 4 according to this comment (see Table 4).

Figures

The presented pathogens in Fig 1-3 based only on male SARI cases with confirmed pathogens or included all pathogens from male and female cases.

Answer: The pathogens in Fig1-3 based on all SARI cases with confirmed pathogens including male and female cases.

---

## [Decision Letter · Decision Letter 1]

17 Feb 2021

PONE-D-20-19561R1

Etiological and Epidemiological Characteristics of Severe Acute Respiratory Infection Caused by Multiple Viruses and Mycoplasma Pneumoniae in Adult Patients in Jinshan, Shanghai: A Pilot Hospital-based Surveillance Study

PLOS ONE

Dear Dr. LI,

Thank you for submitting your manuscript to PLOS ONE. After careful consideration, we feel that it has merit but does not fully meet PLOS ONE’s publication criteria as it currently stands. Therefore, we invite you to submit a revised version of the manuscript that addresses the points raised during the review process.

While the revised manuscript showed significant improvement over previous version, the reviewers still have some concerns that need to be addressed (please see reviewers' insightful comments below).  

In addition, there is still lingering language issue that needs to be addressed (see editor and one of the reviewers PDF files).  We suggest you thoroughly copyedit your manuscript for language usage, spelling, and grammar. If you do not know anyone who can help you do this, you may wish to consider employing a professional scientific editing service.

Whilst you may use any professional scientific editing service of your choice, PLOS has partnered with both American Journal Experts (AJE) and Editage to provide discounted services to PLOS authors. Both organizations have experience helping authors meet PLOS guidelines and can provide language editing, translation, manuscript formatting, and figure formatting to ensure your manuscript meets our submission guidelines. To take advantage of our partnership with AJE, visit the AJE website (http://learn.aje.com/plos/) for a 15% discount off AJE services. To take advantage of our partnership with Editage, visit the Editage website (www.editage.com) and enter referral code PLOSEDIT for a 15% discount off Editage services. If the PLOS editorial team finds any language issues in text that either AJE or Editage has edited, the service provider will re-edit the text for free."

We look forward to receiving your revised manuscript.

Kind regards,

Baochuan Lin, Ph.D.

Academic Editor

PLOS ONE

Reviewers' comments:

Reviewer's Responses to Questions

**Comments to the Author**

1. If the authors have adequately addressed your comments raised in a previous round of review and you feel that this manuscript is now acceptable for publication, you may indicate that here to bypass the “Comments to the Author” section, enter your conflict of interest statement in the “Confidential to Editor” section, and submit your "Accept" recommendation.

Reviewer #1: (No Response)

Reviewer #2: All comments have been addressed

2. Is the manuscript technically sound, and do the data support the conclusions?

Reviewer #1: Yes

Reviewer #2: Yes

3. Has the statistical analysis been performed appropriately and rigorously? 

Reviewer #1: Yes

Reviewer #2: Yes

4. Have the authors made all data underlying the findings in their manuscript fully available?

Reviewer #1: Yes

Reviewer #2: Yes

5. Is the manuscript presented in an intelligible fashion and written in standard English?

Reviewer #1: Yes

Reviewer #2: Yes

6. Review Comments to the Author

Reviewer #1: Thank you for the corrections made, some comments are still remaining

Comment #2: a reference from the WHO is expected as mentioned in the manuscript

Comment #6 :as to ensure reproducibility, you can mention as per the manufacturer's instructions or otherwise specify any other steps not present in the manufacturer's protocol. The reference mentioned does not provide details of the lysis steps

Comment 16#: Further statistical analysis was done for 2 out of the 3 significant P values (remaining is the comorbidity item)

Reviewer #2: Dear Author,

Thank you for making the manuscript beautiful and clean, although minor errors or typos are still present. Please check carefully the attached file and correct these typos.

7. PLOS authors have the option to publish the peer review history of their article (what does this mean?). If published, this will include your full peer review and any attached files.

Reviewer #1: No

Reviewer #2: No

---

## [Author Response · Author response to Decision Letter 1]

2 Mar 2021

Reviewer #1: Thank you for the corrections made, some comments are still remaining

1.Comment #2: a reference from the WHO is expected as mentioned in the manuscript

Answer: We thank for this suggestion. A reference from the WHO has been replaced (see Page 6, line 114 in Revised Manuscript with Track Changes, the same below). 

2.Comment #6 :as to ensure reproducibility, you can mention as per the manufacturer's instructions or otherwise specify any other steps not present in the manufacturer's protocol. The reference mentioned does not provide details of the lysis steps

Answer: This comment was highly appreciated. Actually, this laboratory testing was performed as per the manufacturer’s instructions. We have clarified it (see Page 7, line 141-142).

3.Comment 16#: Further statistical analysis was done for 2 out of the 3 significant P values (remaining is the comorbidity item)

Answer: We are sorry for negligence. The pairwise comparison for comorbidity item has been added (see Page 30, line 560-561).

Response to Reviewer #2' comments: 

Reviewer #2: Dear Author,

Thank you for making the manuscript beautiful and clean, although minor errors or typos are still present. Please check carefully the attached file and correct these typos.

Answer: We highly thank for your carefulness, and we have carefully copyedited the minor errors or typos.

---

## [Editor Report · Decision Letter 2]

5 Mar 2021

Etiological and Epidemiological Characteristics of Severe Acute Respiratory Infection Caused by Multiple Viruses and Mycoplasma Pneumoniae in Adult Patients in Jinshan , Shanghai: A Pilot Hospital-based Surveillance Study

PONE-D-20-19561R2

Dear Dr. LI,

We’re pleased to inform you that your manuscript has been judged scientifically suitable for publication and will be formally accepted for publication once it meets all outstanding technical requirements.

Please correct the following: (1) Delete "(M. pheumoniae)" on line 37 and 90 since it is customary to written out in full the fist time it is ueed in the paper, thereafter, the generic name is abbreviated, and it is not necessary to include the shorten microbial name in parenthesis. (2) Correct "M. pneumonia" to "M. pneumoniae" on line 215 and 241.

Kind regards,

Baochuan Lin, Ph.D.

Academic Editor

PLOS ONE
---

## [Editor Report · Acceptance letter]

12 Mar 2021

PONE-D-20-19561R2 

Etiological and Epidemiological Characteristics of Severe Acute Respiratory Infection Caused by Multiple Viruses and *Mycoplasma Pneumoniae* in Adult Patients in Jinshan, Shanghai: A Pilot Hospital-based Surveillance Study 

Dear Dr. Li:

I'm pleased to inform you that your manuscript has been deemed suitable for publication in PLOS ONE. Congratulations! Your manuscript is now with our production department. 

Kind regards, 

on behalf of

Dr. Baochuan Lin 

Academic Editor

PLOS ONE